# Learning to Share: Simultaneous Parameter Tying and Sparsification in Deep Learning

**Dejiao Zhang**[†][*]
University of Michigan, Ann Arbor, USA
dejiao@umich.edu

**Haozhu Wang**[†]
University of Michigan, Ann Arbor, USA
hzwang@umich.edu

**Mário A.T. Figueiredo**
Instituto de Telecomunicaçõe and Instituto Superior Técnico
University of Lisbon, Portugal
mario.figueiredo@lx.it.pt

**Laura Balzano**[*]
University of Michigan, Ann Arbor, USA
girasole@umich.edu

## Abstract

*Deep neural networks* (DNNs) may contain millions, even billions, of parameters/weights, making storage and computation very expensive and motivating a large body of work aimed at reducing their complexity by using, e.g., sparsity-inducing regularization. Parameter sharing/tying is another well-known approach for controlling the complexity of DNNs by forcing certain sets of weights to share a common value. Some forms of weight sharing are hard-wired to express certain invariances; a notable example is the shift-invariance of convolutional layers. However, other groups of weights may be tied together during the learning process to further reduce the network complexity. In this paper, we adopt a recently proposed regularizer, GrOWL (*group ordered weighted* $\ell_1$), which encourages sparsity and, simultaneously, learns which groups of parameters should share a common value. GrOWL has been proven effective in linear regression, being able to identify and cope with strongly correlated covariates. Unlike standard sparsity-inducing regularizers (*e.g.*, $\ell_1$ a.k.a. Lasso), GrOWL not only eliminates unimportant neurons by setting all their weights to zero, but also explicitly identifies strongly correlated neurons by tying the corresponding weights to a common value. This ability of GrOWL motivates the following two-stage procedure: (i) use GrOWL regularization during training to simultaneously identify significant neurons and groups of parameters that should be tied together; (ii) retrain the network, enforcing the structure that was unveiled in the previous phase, *i.e.*, keeping only the significant neurons and enforcing the learned tying structure. We evaluate this approach on several benchmark datasets, showing that it can dramatically compress the network with slight or even no loss on generalization accuracy.

## 1 Introduction

*Deep neural networks* (DNNs) have recently revolutionized machine learning by dramatically advancing the state-of-the-art in several applications, ranging from speech and image recognition to playing video games (Goodfellow et al., 2016). A typical DNN consists of a sequence of concatenated layers, potentially involving millions or billions of parameters; by using very large training sets, DNNs are able to learn extremely complex non-linear mappings, features, and dependencies.

A large amount of research has focused on the use of regularization in DNN learning (Goodfellow et al., 2016), as a means of reducing the generalization error. It has been shown that the parametrization of many DNNs is very redundant, with a large fraction of the parameters being predictable from the remaining ones, with no accuracy loss (Denil et al., 2013). Several regularization methods have been proposed to tackle the potential over-fitting due to this redundancy. Arguably, the earliest

---

[*]Both Dejiao Zhang and Laura Balzano's participations were funded by DARPA-16-43-D3M-FP-037.

[†]Co-first author.

and simplest choice is the classical $\ell_2$ norm, known as *weight decay* in the early neural networks literature (Rumelhart et al., 1986), and as *ridge regression* in statistics. In the past two decades, sparsity-inducing regularization based on the $\ell_1$ norm (often known as Lasso) (Tibshirani, 1996), and variants thereof, became standard tools in statistics and machine learning, including in deep learning (Goodfellow et al., 2016). Recently, Scardapane et al. (2017) used group-Lasso (a variant of Lasso that assumes that parameters are organized in groups and encourages sparsity at the group level (Yuan & Lin, 2006)) in deep learning. One of the effects of Lasso or group-Lasso regularization in learning a DNN is that many of the parameters may become exactly zero, thus reducing the amount of memory needed to store the model, and lowering the computational cost of applying it.

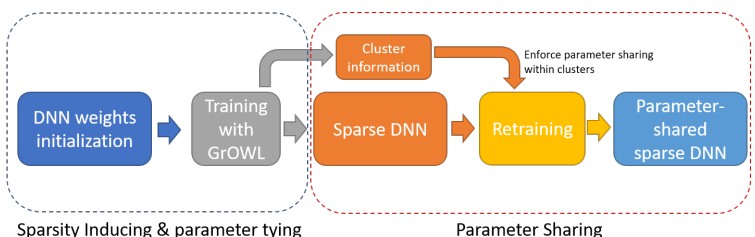

Figure 1: A DNN is first trained with GrOWL regularization to simultaneously identify the sparse but significant connectivities and the correlated cluster information of the selected features. We then retrain the neural network only in terms of the selected connectivities while enforcing parameter sharing within each cluster.

It has been pointed out by several authors that a major drawback of Lasso (or group-Lasso) regularization is that in the presence of groups of highly correlated covariates/features, it tends to select only one or an arbitrary convex combination of features from each group (Bondell & Reich, 2008; Bühlmann et al., 2013; Figueiredo & Nowak, 2016; Oswal et al., 2016; Zou & Hastie, 2005). Moreover, the learning process tends to be unstable, in the sense that subsets of parameters that end up being selected may change dramatically with minor changes in the data or algorithmic procedure. In DNNs, it is almost unavoidable to encounter correlated features, not only due to the high dimensionality of the input to each layer, but also because neurons tend to co-adapt, yielding strongly correlated features that are passed as input to the subsequent layer (Srivastava et al., 2014).

In this work, we propose using, as a regularizer for learning DNNs, the group version of the *ordered weighted* $\ell_1$ (OWL) norm (Figueiredo & Nowak, 2016), termed group-OWL (GrOWL), which was recently proposed by Oswal et al. (2016). In a linear regression context, GrOWL regularization has been shown to avoid the above mentioned deficiency of group-Lasso regularization. In addition to being a sparsity-inducing regularizer, GrOWL is able to explicitly identify groups of correlated features and set the corresponding parameters/weights to be very close or exactly equal to each other, thus taking advantage of correlated features, rather than being negatively affected by them. In deep learning parlance, this corresponds to adaptive *parameter sharing/tying*, where instead of having to define *a priori* which sets of parameters are forced to share a common value, these sets are learned during the training process. We exploit this ability of GrOWL regularization to encourage parameter sparsity and group-clustering in a two-stage procedure depicted in Fig. 1: we first use GrOWL to identify the significant parameters/weights of the network and, simultaneously, the correlated cluster information of the selected features; then, we retrain the network only in terms of the selected features, while enforcing the weights within the same cluster to share a common value.

The experiments reported below confirm that using GrOWL regularization in learning DNNs encourages sparsity and also yields parameter sharing, by forcing groups of weights to share a common absolute value. We test the proposed approach on two benchmark datasets, MNIST and CIFAR-10, comparing it with weight decay and group-Lasso regularization, and exploring the accuracy-memory trade-off. Our results indicate that GrOWL is able to reduce the number of free parameters in the network without degrading the accuracy, as compared to other approaches.

## 2 RELATED WORK

In order to relieve the burden on both required memory and data for training and storing DNNs, a substantial amount of work has focused on reducing the number of free parameters to be estimated,

namely by enforcing weight sharing. The classical instance of sharing is found in the convolutional layers of DNNs (Goodfellow et al., 2016). In fact, weight-sharing as a simplifying technique for NNs can be traced back to more than 30 years ago (LeCun, 1987; Rumelhart & McClelland, 1986).

Recently, there has been a surge of interest in compressing the description of DNNs, with the aim of reducing their storage and communication costs. Various methods have been proposed to approximate or quantize the learned weights after the training process. Denton et al. (2014) have shown that, in some cases, it is possible to replace the original weight matrix with a low-rank approximation. Alternatively, Aghasi et al. (2016) propose retraining the network layer by layer, keeping the layer inputs and outputs close to the originally trained model, while seeking a sparse transform matrix, whereas Gong et al. (2014) propose using vector quantization to compress the parameters of DNNs.

Network pruning is another relevant line of work. In early work, LeCun et al. (1989) and Hassibi & Stork (1993) use the information provided by the Hessian of the loss function to remove less important weights; however, this requires expensive computation of second order derivatives. Recently, Han et al. (2016) reduce the number of parameters by up to an order of magnitude by alternating between learning the parameters and removing those below a certain threshold. Li et al. (2016) propose to prune filters, which seeks sparsity with respect to neurons, rather than connections; that approach relieves the burden on requiring sparse libraries or special hardware to deploy the network. All those methods either require multiple training/retraining iterations or a careful choice of thresholds.

There is a large body of work on sparsity-inducing regularization in deep learning. For example, Collins & Kohli (2014) exploit $\ell_1$ and $\ell_0$ regularization to encourage weight sparsity; however, the sparsity level achieved is typically modest, making that approach not competitive for DNN compression. Group-Lasso has also been used in training DNNs; it allows seeking sparsity in terms of neurons (Scardapane et al., 2017; Alvarez & Salzmann, 2016; Zhou et al., 2016; Murray & Chiang, 2015) or other structures, *e.g.*, filters, channels, filter shapes, and layer depth (Wen et al., 2016). However, as mentioned above, both Lasso and group-Lasso can fail in the presence of strongly correlated features (as illustrated in Section 4, with both synthetic data and real data.

A recent stream of work has focused on using further parameter sharing in convolutional DNNs. By tying weights in an appropriate way, Dieleman et al. (2016) obtain a convolutional DNN with rotation invariance. On the task of analyzing positions in the game *Go*, Clark & Storkey (2015) showed improved performance by constraining features to be invariant to reflections along the x-axis, y-axis, and diagonal-axis. Finally, Chen et al. (2015) used a hash function to randomly group the weights such that those in a hash bucket share the same value. In contrast, with GrOWL regularization, we aim to learn weight sharing from the data itself, rather than specifying it *a priori*.

*Dropout*-type methods have been proposed to fight over-fitting and are very popular, arguably due to their simplicity of implementation (Srivastava et al., 2014). Dropout has been shown to effectively reduce over-fitting and prevent different neurons from co-adapting. Decorrelation is another popular technique in deep learning pipelines (Bengio & Bergstra, 2009; Cogswell et al., 2015; Rodríguez et al., 2016); unlike sparsity-inducing regularizers, these methods try to make full use of the model's capacity by decorrelating the neurons. Although dropout and decorrelation can reduce over-fitting, they do not compress the network, hence do not address the issue of high memory cost. It should also be mentioned that our proposal can be seen as complementary to dropout and decorrelation: whereas dropout and decorrelation can reduce co-adaption of nodes during training, GrOWL regularization copes with co-adaptation by tying together the weights associated to co-adapted nodes.

## 3 Group-OWL Regularization for Deep Learning

### 3.1 The Group-OWL Norm

We start by recalling the definition of the group-OWL (GrOWL) regularizer and very briefly reviewing some of its relevant properties (Oswal et al., 2016).

**Definition 1.** *Given a matrix $W \in \mathbb{R}^{n \times m}$, let $w_{[i].}$ denote the row of $W$ with the $i$-th largest $\ell_2$ norm. Let $\lambda \in \mathbb{R}_+^n$, with $0 < \lambda_1 \geq \lambda_2 \geq \cdots \geq \lambda_n \geq 0$. The GrOWL regularizer (which is a norm) $\Omega_\lambda : \mathbb{R}^{n \times m} \to \mathbb{R}$ is defined as*

$$\Omega_\lambda(W) = \sum_{i=1}^{n} \lambda_i \left\| w_{[i].} \right\| \tag{1}$$

This is a group version of the OWL regularizer (Figueiredo & Nowak, 2016), also known as WSL1 (*weighted sorted $\ell$1* (Zeng & Figueiredo, 2014)) and SLOPE (Bogdan et al., 2015), where the groups are the rows of its matrix argument. It is clear that GrOWL includes group-Lasso as a special case when $\lambda_1 = \lambda_n$. As a regularizer for multiple/multi-task linear regression, each row of $W$ contains the regression coefficients of a given feature, for the $m$ tasks. It has been shown that by adding the GrOWL regularizer to a standard squared-error loss function, the resulting estimate of $W$ has the following property: rows associated with highly correlated covariates are very close or even exactly equal to each other (Oswal et al., 2016). In the linear case, GrOWL encourages correlated features to form predictive clusters corresponding to the groups of rows that are nearly or exactly equal. The rationale underlying this paper is that when used as a regularizer for DNN learning, GrOWL will induce both sparsity and parameters tying, as illustrated in Fig. 2 and explained below in detail.

### 3.2 Layer-Wise GrOWL Regularization For Feedforward Neural Networks

A typical feed-forward DNN with $L$ layers can be treated as a function $f$ of the following form:

$$f(x,\theta) \equiv h_L = f_L\left(h_{L-1}W_L + b_L\right), h_{L-1} = f_{L-1}\left(h_{L-2}W_{L-1} + b_{L-1}\right), \ldots, h_1 = f_1\left(xW_1 + b_1\right)$$

$\theta = \left(W_1, b_1, \ldots, W_L, b_L\right)$ denotes the set of parameters of the network, and each $f_i$ is a component-wise nonlinear activation function, with the *rectified linear unit* (ReLU), the *sigmoid*, and the *hyperbolic tangent* being common choices for this function (Goodfellow et al., 2016).

Given labelled data $\mathcal{D} = \left((x^{(1)}, y^{(1)}), ..., (x^{(m)}, y^{(m)})\right)$, DNN learning may be formalized as an optimization problem,

$$\min_{\theta} \mathcal{L}(\theta) + \mathcal{R}(\theta), \quad \text{with} \quad \mathcal{L}(\theta) = \sum_{i=1}^{m} L\left(y^{(i)}, f\left(x^{(i)}, \theta\right)\right) , \tag{2}$$

where $L\left(y, \hat{y}\right)$ is the loss incurred when the DNN predicts $\hat{y}$ for $y$, and $\mathcal{R}$ is a regularizer. Here, we adopt as regularizer a sum of GrOWL penalties, each for each layer of the neural network, *i.e.,*

$$\mathcal{R}(\theta) = \sum_{l=1}^{L} \Omega_{\lambda^{(l)}}\left(W_l\right), \quad \lambda^{(l)} \in \mathbb{R}_+^{N_{l-1}} , \tag{3}$$

where $N_l$ denotes the number of neurons in the $l$-th layer and $0 < \lambda_1^{(l)} \geq \lambda_2^{(l)} \geq \cdots \geq \lambda_{N_{l-1}}^{(l)} \geq 0$. Since $\mathcal{R}(\theta)$ does not depend on $b_1, ..., b_L$, the biases are not regularized, as is common practice.

As indicated in Eq. (3), the number of groups in each GrOWL regularizer is the number of neurons in the previous layer, *i.e.,* $\lambda^{(l)} \in \mathbb{R}^{N_{l-1}}$. In other words, *we treat the weights associated with each input feature as a group*. For fully connected layers, where $W_l \in \mathbb{R}^{N_{l-1} \times N_l}$, each group is a row of the weight matrix. In convolutional layers, where $W_l \in \mathbb{R}^{F_w \times F_h \times N_{l-1} \times N_l}$, with $F_w$ and $F_h$ denoting the width and height, respectively, of each filter, we first reshape $W_l$ to a 2-dimensional array, *i.e.,* $W_l \to W_l^{2D}$, where $W_l^{2D} \in \mathbb{R}^{N_{l-1} \times (F_w F_h N_l)}$, and then apply GrOWL on the reshaped matrix. That is, if the $l$-th layer is convolutional, then

$$\mathcal{R}(W_l) = \Omega_{\lambda^{(l)}}\left(W_l^{2D}\right). \tag{4}$$

Each row of $W_l^{2D}$ represents the operation on an input channel. The rationale to apply the GrOWL regularizer to each row of the reshaped weight matrix is that GrOWL can select the relevant features of the network, while encouraging the coefficient rows of each layer associated with strongly correlated features from the previous layer to be nearly or exactly equal, as depicted in Fig. 2. The goal is to significantly reduce the complexity by: **(i)** pruning unimportant neurons of the previous layer that correspond to zero rows of the (reshaped) weight matrix of the current layer; **(ii)** grouping the rows associated with highly correlated features of the previous layer, thus encouraging the coefficient rows in each of these groups to be very close to each other. As a consequence, in the retraining process, we can further compress the neural network by enforcing the parameters *within each neuron* that belong to the same cluster to share same values.

In the work of Alvarez & Salzmann (2016), each group is predefined as the set of parameters associated to a neuron, and group-Lasso regularization is applied to seek group sparsity, which corresponds to zeroing out redundant neurons of each layer. In contrast, we treat the filters corresponding

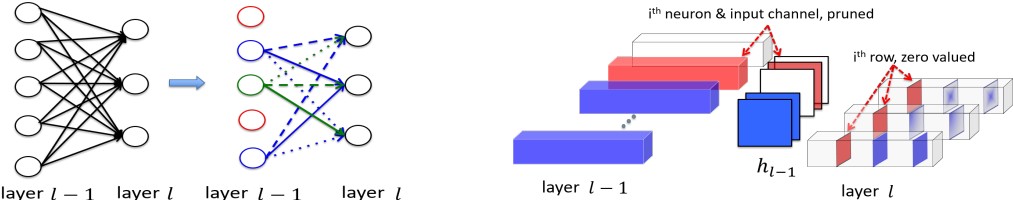

Figure 2: GrOWL's regularization effect on DNNs. **Fully connected layers (Left):** for layer $l$, GrOWL clusters the input features from the previous layer, $l-1$, into different groups, *e.g.,* blue and green. *Within each neuron* of layer $l$, the weights associated with the input features from the same cluster (input arrows marked with the same color) share the same parameter value. The neurons in layer $l-1$ corresponding to zero-valued rows of $W_l$ have zero input to layer $l$, hence get removed automatically. **Convolutional layers (right):** each group (row) is predefined as the filters associated with the same input channel; parameter sharing is enforced among the filters within each neuron that corresponds with the same cluster (marked as blue with different effects) of input channels.

to the same input channel as a group, and GrOWL is applied to prune the redundant groups and thus *remove the associated unimportant neurons of the previous layer*, while grouping associated parameters of the current layer that correspond with highly correlated input features to different clusters. Moreover, as shown in Section 4, group-Lasso can fail at selecting all relevant features of previous layers, and for the selected ones the corresponding coefficient groups are quite dissimilar from each other, making it impossible to further compress the DNN by enforcing parameter tying.

### 3.3 PROXIMAL GRADIENT ALGORITHM

To solve (2), we use a *proximal gradient algorithm* (Bauschke & Combettes, 2011), which has the following general form: at the $t$-th iteration, the parameter estimates are updated according to

$$\theta^{(t+1)} = \text{prox}_{\eta\mathcal{R}}\left(\theta^{(t)} - \eta\nabla_\theta\mathcal{L}(\theta^{(t)})\right), \tag{5}$$

where, for some convex function $Q$, $\text{prox}_Q$ denotes its *proximity operator* (or simply "prox") (Bauschke & Combettes, 2011), defined as $\text{prox}_Q(\xi) = \arg\min_\nu Q(\nu) + \frac{1}{2}\|\nu-\xi\|_2^2$. In Eq. (5), $\|\nu-\xi\|_2^2$ denotes the sum of the squares of the differences between the corresponding components of $\nu$ and $\xi$, regardless of their organization (here, a collection of matrices and vectors).

Since $\mathcal{R}(\theta)$, as defined in (3), is separable across the weight matrices of different layers and zero for $b_1, ..., b_L$, the corresponding prox is also separable, thus

$$W_l^{(t+1)} = \text{prox}_{\eta\Omega_\lambda^{(l)}}\left(W_l^{(t)} - \eta\,\nabla_{W_l}\mathcal{L}(\theta^{(t)})\right), \quad \text{for } l = 1,\dots,L \tag{6}$$

$$b_l^{(t+1)} = b_l^{(t)} - \eta\,\nabla_{b_l}\mathcal{L}(\theta^{(t)}) \quad \text{for } l = 1, ..., L. \tag{7}$$

It was shown by Oswal et al. (2016) that the prox of GrOWL can be computed as follows. For some matrix $V \in \mathbb{R}^{N\times M}$, let $U = \text{prox}_{\Omega_\lambda}(V)$, and $v_i$ and $u_i$ denote the corresponding $i$-th rows. Then,

$$u_i = v_i\,(\text{prox}_{\Omega_\lambda}(\tilde{v}))_i/\|v_i\|, \tag{8}$$

where $\tilde{v} = [\|v_1\|, \|v_2\|, \cdots, \|v_N\|]$. For vectors in $\mathbb{R}^N$ (in which case GrOWL coincides with OWL), $\text{prox}_{\Omega_{\lambda^{(l)}}}$ can be computed with $O(n\log n)$ cost, where the core computation is the so-called *pool adjacent violators algorithm* (PAVA (de Leeuw et al., 2009)) for isotonic regression. We provide one of the existing algorithms in Appendix A; for details, the reader is referred to the work of Bogdan et al. (2015) and Zeng & Figueiredo (2015). In this paper, we apply the proximal gradient algorithm per epoch, which generally performs better. The training method is summarized in Algorithm 1.

---

**Algorithm 1**

---

**Input**: parameters of the OWL regularizers $\lambda^{(l)}, ..., \lambda^{(L)}$, learning rate $\eta$
**for** each epoch $T$ **do**
    **for** each iteration t in epoch T **do**
        Update the parameters $\theta = \left(W_1, b_1, \ldots, W_L, b_L\right)$ via backpropagation (BP)
    **end for**
    Apply proximity operator via (6)
**end for**

---

## 3.4 IMPLEMENTATION DETAILS

### 3.4.1 SETTING THE GROWL WEIGHTS

GrOWL is a family of regularizers, with different variants obtained by choosing different weight sequences $\lambda_1, \ldots, \lambda_n$. In this paper, we propose the following choice:

$$\lambda_i = \begin{cases} \Lambda_1 + (p - i + 1)\Lambda_2, & \text{for } i = 1, ..., p, \\ \Lambda_1, & \text{for } i = p + 1, ..., n, \end{cases} \qquad (9)$$

where $p \in \{1, ...n\}$ is a parameter. The first $p$ weights follow a linear decay, while the remaining ones are all equal to $\Lambda_1$. Notice that, if $p = n$, the above setting is equivalent to OSCAR (Bondell & Reich, 2008). Roughly speaking, $\Lambda_1$ controls the sparsifying strength of the regularizer, while $\Lambda_2$ controls the clustering property (correlation identification ability) of GrOWL (Oswal et al., 2016). Moreover, by setting the weights to a common constant beyond index $p$ means that clustering is only encouraged among the $p$ largest coefficients, *i.e.,* only among relevant coefficient groups.

Finding adequate choices for $p$, $\Lambda_1$, and $\Lambda_2$ is crucial for jointly selecting the relevant features and identifying the underlying correlations. In practice, we find that with properly chosen $p$, GrOWL is able to find more correlations than OSCAR. We explore different choices of $p$ in Section 4.1.

### 3.4.2 PARAMETER TYING

After the initial training phase, at each layer $l$, rows of $W_l$ that corresponds to highly correlated outputs of layer $l - 1$ have been made similar or even exactly equal. To further compress the DNN, we force rows that are close to each other to be identical. We first group the rows into different clusters [1] according to the *pairwise similarity* metric

$$\mathcal{S}_l(i, j) = \frac{W_{l,i}^T W_{l,j}}{\max\left(\|W_{l,i}\|_2^2, \|W_{l,j}\|_2^2\right)} \in [-1, 1], \qquad (10)$$

where $W_{l,i}$ and $W_{l,j}$ denote the $i$-th and $j$-th rows of $W_l$, respectively.

With the cluster information obtained by using GrOWL, we enforce parameter sharing for the rows that belong to a same cluster by replacing their values with the averages (centroid) of the rows in that cluster. In the subsequent retraining process , let $\mathcal{G}_k^{(l)}$ denote the $k$-th cluster of the $l$-th layer, then centroid $g_k^{(l)}$ of this cluster is updated via

$$\frac{\partial \mathcal{L}}{\partial g_k^{(l)}} = \frac{1}{\left|\mathcal{G}_k^{(l)}\right|} \sum_{W_{l,i} \in \mathcal{G}_k^{(l)}} \frac{\partial \mathcal{L}}{\partial W_{l,i}}. \qquad (11)$$

## 4 NUMERICAL RESULTS

We assess the performance of the proposed method on two benchmark datasets: MNIST and CIFAR-10. We consider two different networks and compare GrOWL with group-Lasso and weight decay, in terms of the compression vs accuracy trade-off. For fair comparison, the

---

[1]In this paper, we use the built-in *affinity propagation* method of the scikit-learn package (Buitinck et al., 2013). A brief description of the algorithm is provided in Appendix B.

training-retraining pipeline is used with the different regularizers. After the initial training phase, the rows that are close to each other are clustered together and forced to share common values in the retraining phase. We implement all models using Tensorflow Abadi et al. (2016). We evaluate the effect of the different regularizers using the following quantities: sparsity = (#zero params)/(# total params), compression rate = (# total params)/(# unique params), and parameter sharing = (# nonzero params)/(# unique params).

## 4.1 DIFFERENT CHOICES OF GROWL PARAMETERS

First, we consider a synthetic data matrix $X$ with block-diagonal covariance matrix $\Sigma$, where each block corresponds to a cluster of correlated features, and there is a gap $g$ between two blocks. Within each cluster, the covariance between two features $X_i$ and $X_j$ is $\text{cov}(X_i, X_j) = 0.96^{|i-j|}$, while features from different clusters are generated independently of each other. We set $n = 784, K = 10$, block size 50, and gap $g = 28$. We generate 10000 training and 1000 testing examples.

We train a NN with a single fully-connected layer of 300 hidden units. Fig 3 shows the first 25000 entries of the sorted *pairwise similarity* matrices (Eq 10) obtained by applying GrOWL with different $p$ (Eq 9) values. By setting the weights beyond index $p$ to a common constant implies that clustering is only encouraged among the $p$ largest coefficients, *i.e.*, relevant coefficient groups; however, Fig. 3 shows that, with properly chosen $p$, GrOWL yields more parameter tying than OSCAR ($p = n$). On the other hand, smaller $p$ values allow using large $\Lambda_2$, encouraging parameter tying among relatively loose correlations. In practice, we find that for $p$ around the target fraction of nonzero parameters leads to good performance in general. The intuition is that we only need to identify correlations among the selected important features.

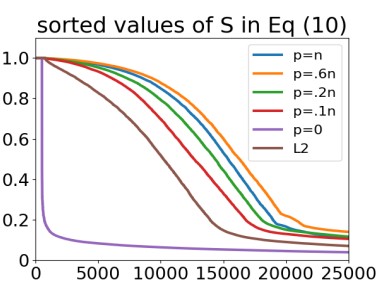

Figure 3: Regularization effect of GrOWL for different p values (Eq. (9)).

Fig. 3 shows that weight decay (denoted as $\ell_2$) also pushes parameters together, though the parameter-tying effect is not as clear as that of GrOWL. As has been observed in the literature (Bondell & Reich, 2008), weight decay often achieves better generalization than sparsity-inducing regularizers. It achieves this via parameter shrinkage, especially in the highly correlated region, but it does not yield sparse models. In the following section, we explore the compression performance of GrOWL by comparing it with both group-Lasso and weight decay. We also explore how to further improve the accuracy vs compression trade-off by using sparsity-inducing regularization together with weight decay ($\ell_2$). For each case, the baseline performance is provided as the best performance obtained by running the original neural network (without compression) after sweeping the hyper-parameter on the weight decay regularizer over a range of values.

## 4.2 FULLY CONNECTED NEURAL NETWORK ON MNIST

The MNIST dataset contains centered images of handwritten digits (0–9), of size 28×28 (784) pixels. Fig 4 (a) shows the (784 × 784) correlation matrix of the dataset (the margins are zero due to the redundant background of the images). We use a network with a single fully connected layer of 300 hidden units. The network is trained for 300 epochs and then retrained for an additional 100 epochs, both with momentum. The initial learning rate is set to 0.001, for both training and retraining, and is reduced by a factor of 0.96 every 10 epochs. We set $p = 0.5$, and $\Lambda_1, \Lambda_2$ are selected by grid search.

Pairwise similarities (see Eq. (10)) between the rows of the weight matrices learned with different regularizers are shown in Fig. 4 (b–f). As we can see, GrOWL ($+\ell_2$) identifies more correlations than group-Lasso ($+\ell_2$), and the similarity patterns in Fig. 4 (b, c) are very close to that of the data (Fig. 4(a)). On the other hand, weight decay also identifies correlations between parameter rows, but it does not induce sparsity. Moreover, as shown in Table 1, GrOWL yields a higher level of parameter sharing than weight decay, matching what we observed on synthetic data in Section 4.1.

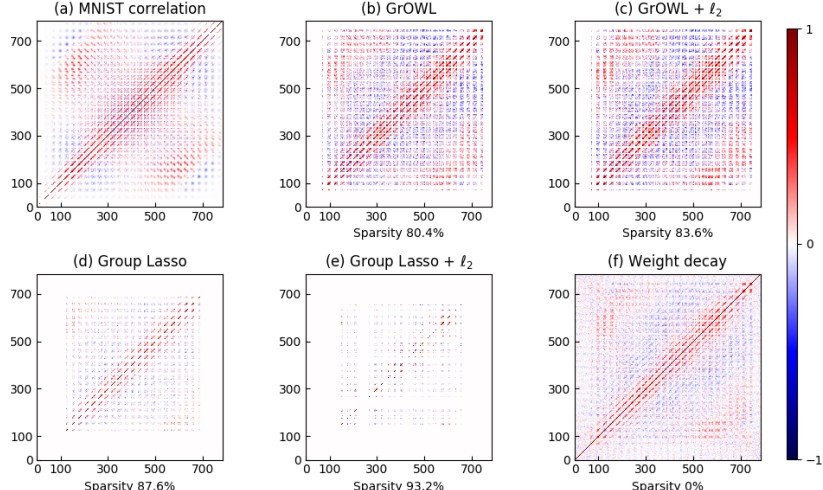

Figure 4: MNIST: comparison of the data correlation and the pairwise similarity maps (Eq (10)) of the parameter rows obtained by training the neural network with GrOWL, GrOWL+$\ell_2$, group-Lasso, group-Lasso+$\ell_2$ and weight decay ($\ell_2$).

Table 1: Sparsity, parameter sharing, and compression rate results on MNIST. Baseline model is trained with weight decay and we do not enforce parameter sharing for baseline model. We train each model for 5 times and report the average values together with their standard deviations.

| Regularizer | Sparsity | Parameter Sharing | Compression ratio | Accuracy |
|---|---|---|---|---|
| none | $0.0 \pm 0\%$ | $1.0 \pm 0$ | $1.0 \pm 0$ | $98.3 \pm 0.1\%$ |
| weight decay | $0.0 \pm 0\%$ | $1.6 \pm 0$ | $1.6 \pm 0$ | $98.4 \pm 0.0\%$ |
| group-Lasso | $87.6 \pm 0.1\%$ | $1.9 \pm 0.1$ | $15.8 \pm 1.0$ | $98.1 \pm 0.1\%$ |
| group-Lasso+$\ell_2$ | $93.2 \pm 0.4\%$ | $1.6 \pm 0.1$ | $23.7 \pm 2.1$ | $98.0 \pm 0.1\%$ |
| GrOWL | $80.4 \pm 1.0\%$ | $3.2 \pm 0.1$ | $16.7 \pm 1.3$ | $98.1 \pm 0.1\%$ |
| GrOWL+$\ell_2$ | $83.6 \pm 0.5\%$ | $3.9 \pm 0.1$ | $24.1 \pm 0.8$ | $98.1 \pm 0.1\%$ |

The compression vs accuracy trade-off of the different regularizers is summarized in Table 1, where we see that applying $\ell_2$ regularization together with group-Lasso or GrOWL leads to a higher compression ratio, with negligible effect on the accuracy. Table 1 also shows that, even with lower sparsity after the initial training phase, GrOWL ($+\ell_2$) compresses the network more than group-Lasso ($+\ell_2$), due to the significant amount of correlation it identifies; this also implies that group-Lasso only selects a subset of the correlated features, while GrOWL selects all of them. On the other hand, group-Lasso suffers from randomly selecting a subset of correlated features; this effect is illustrated in Fig. 5, which plots the indices of nonzero rows, showing that GrOWL ($+\ell_2$) stably selects relevant features while group-Lasso ($+\ell_2$) does not. The mean ratios of changed indices[2] are 11.09%, 0.59%, 32.07%, and 0.62% for group-Lasso, GrOWL, group-Lasso+$\ell_2$, and GrOWL+$\ell_2$, respectively.

### 4.3 VGG-16 ON CIFAR-10

To evaluate the proposed method on large DNNs, we consider a VGG-like (Simonyan & Zisserman, 2014) architecture proposed by Zagoruyko (2015) on the CIFAR-10 dataset. The network architecture is summarized in Appendix C; comparing with the original VGG of Simonyan & Zisserman (2014), their fully connected layers are replaced with two much smaller ones. A batch normalization layer is added after each convolutional layer and the first fully connected layer. Unlike Zagoruyko (2015), we don't use dropout. We first train the network under different regularizers for 150 epochs, then retrain it for another 50 epochs, using the learning rate decay scheme described by He et al.

---

[2]The mean ratio of changed indices is defined as: $\frac{1}{n} \sum_{k=1}^{n} \|I_k - \bar{I}\|_0 / \|\bar{I}\|_0$, where $n$ is the number of experiments, $I_k$ is the index vector of $k$th experiment, and $\bar{I} = \frac{1}{n} \sum_{k=1}^{n} I_k$ is the mean index vector.

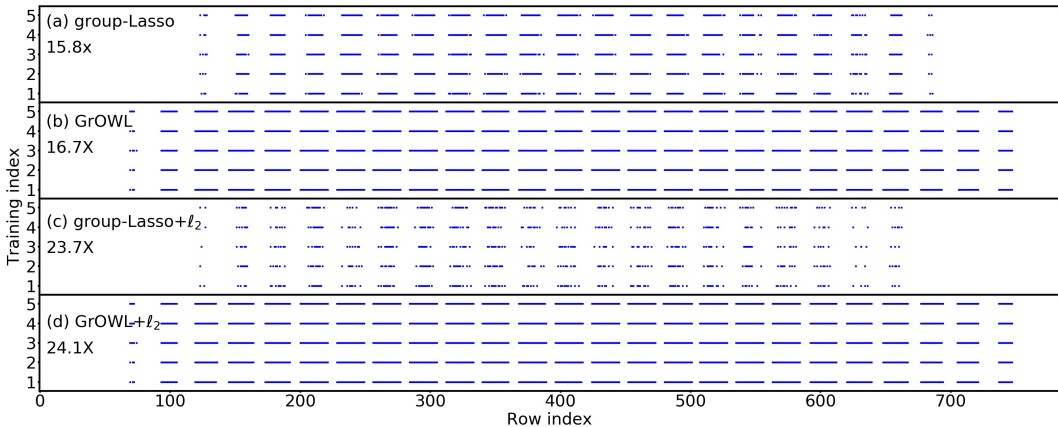

Figure 5: MNIST: sparsity pattern of the trained fully connected layer, for 5 training runs, using group-Lasso, GrOWL, group-Lasso+$\ell_2$, GrOWL+$\ell_2$.

Table 2: Sparsity (**S1**) and Parameter Sharing (**S2**) of VGG-16 on CIFAR-10. Layers marked by * are regularized. We report the averaged results over 5 runs.

| Layers | Weight Decay (**S1**, **S2**) | group-Lasso (**S1**, **S2**) | group-Lasso + $\ell_2$ (**S1**, **S2**) | GrOWL (**S1**, **S2**) | GrOWL + $\ell_2$ (**S1**, **S2**) |
|---|---|---|---|---|---|
| conv1 | 0%, 1.0 | 0%, 1.0 | 0%, 1.0 | 0%,1.0 | 0%, 1.0 |
| *conv2 | 0%, 1.0 | 34%, 1.0 | 40%, 1.0 | 20%, 1.0 | 34%, 1.0 |
| *conv3 | 0%, 1.0 | 28%, 1.0 | 20%, 1.0 | 28%, 1.0 | 17%, 1.0 |
| *conv4 | 0%, 1.0 | 34%, 1.0 | 29%, 1.0 | 30%, 1.0 | 27% 1.0 |
| *conv5 | 0%, 1.0 | 12%, 1.0 | 11%, 1.0 | 8%, 1.0 | 14%, 1.0 |
| *conv6 | 0%, 1.0 | 38%, 1.0 | 40%, 1.0 | 38%, 1.0 | 43%, 1.0 |
| *conv7 | 0%, 1.0 | 46%, 1.0 | 51%, 1.0 | 40%, 1.0 | 50%, 1.0 |
| *conv8 | 0%, 1.0 | 49%, 1.0 | 53%, 1.0 | 50%, 1.0 | 55%, 1.0 |
| *conv9 | 0%, 1.0 | 78%, 1.0 | 78%, 1.0 | 74%, 1.1 | 75%, 1.2 |
| *conv10 | 0%, 1.2 | 76%, 1.0 | 76%, 1.0 | 66%, 2.7 | 73%, 3.0 |
| *conv11 | 0%, 1.2 | 84%, 1.0 | 87%, 1.0 | 81%, 3.7 | 88%, 3.7 |
| *conv12 | 0%, 2.0 | 85%, 1.0 | 91%, 1.0 | 75%, 2.6 | 78%, 2.5 |
| *conv13 | 0%, 2.1 | 75%, 1.1 | 90%, 1.1 | 78%, 1.9 | 71%, 4.2 |
| *fc | 0%, 4.2 | 78%, 1.0 | 91%, 1.1 | 69%, 2.7 | 81%, 2.2 |
| softmax | 0%, 1.0 | 0%,1.0 | 0%, 1.0 | 0%, 1.0 | 0%, 1.0 |
| Compression | $1.3 \pm 0.1$X | $11.1 \pm 0.5$X | $14.5 \pm 0.5$X | $11.4 \pm 0.5$X | $14.5 \pm 0.5$X |
| Accuracy | $93.1 \pm 0.0\%$ | $92.1 \pm 0.2\%$ | $92.7 \pm 0.1\%$ | $92.2 \pm 0.1\%$ | $92.7 \pm 0.1\%$ |
| Baseline | Accuracy: $93.4 \pm 0.2\%$, Compression: 1.0X | | | | |

(2016): the initial rates for the training and retraining phases are set to $0.01$ and $0.001$, respectively; the learning rate is multiplied by $0.1$ every 60 epochs of the training phase, and every 20 epochs of the retraining phase. For GrOWL ($+\ell_2$), we set $p = 0.1\,n$ (see Eq. (9)) for all layers, where $n$ denotes the number of rows of the (reshaped) weight matrices of each layer.

The results are summarized in Table 2. For all of the regularizers, we use the *affinity propagation* algorithm (with preference value[3] set to $0.8$) to cluster the rows at the end of initial training process. Our experiments showed that it is hard to encourage parameter tying in the first 7 convolutional layers; this may be because the filters of these first 7 convolutional layers have comparatively large feature maps (from $32 \times 32$ to $8 \times 8$), which are only loosely correlated. We illustrate this reasoning in Fig. 6, showing the cosine similarity between the vectorized output channels of layers 1, 6, 10, and 11, at the end of the training phase; it can be seen that the outputs of layers 10 and 11 have many more significant similarities than that of layer 6. Although the output channels of layer 1 also

---

[3]In Table 4 (Appendix C), we explore the effect of different choices of this value.

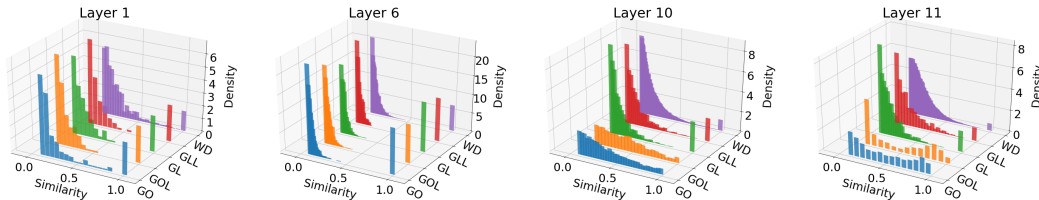

Figure 6: Output channel cosine similarity histogram obtained with different regularizers. Labels: **GO**:GrOWL, **GOL**:GrOWL+$\ell_2$, **GL**:group-Lasso, **GLL**:group-Lasso+$\ell_2$, **WD**:weight decay.

have certain similarities, as seen in Table 2, neither GrOWL ($+\ell_2$) nor weight decay tends to tie the associated weights. This may mean that the network is maintaining the diversity of the inputs in the first few convolutional layers.

Although GrOWL and weight decay both encourage parameter tying in layers 9-13, weight decay does it with less intensity and does not yield a sparse model, thus it cannot significantly compress the network. Li et al. (2016) propose to prune small weights after the initial training phase with weight decay, then retrain the reduced network; however, this type of method only achieves compression[4] ratios around 3. As mentioned by Li et al. (2016), layers 3-7 can be very sensitive to pruning; however, both GrOWL ($+\ell_2$) and group-Lasso ($+\ell_2$) effectively compress them, with minor accuracy loss.

On the other hand, similar to what we observed by running the simple fully-connected network on MNIST, the accuracy-memory trade-off improves significantly by applying GrOWL or group-Lasso together with $\ell_2$. However, Table 2 also shows that the trade-off achieved by GrOWL ($+\ell_2$) and group-Lasso ($+\ell_2$) are almost the same. We suspect that this is caused by the fact that CIFAR-10 is simple enough that one could still expect a good performance after strong network compression. We believe this gap in the compression vs accuracy trade-off can be further increased in larger networks on more complex datasets. We leave this question for future research.

## 5 CONCLUSION

We have proposed using the recent GrOWL regularizer for simultaneous parameter sparsity and tying in DNN learning. By leveraging on GrOWL's capability of simultaneously pruning redundant parameters and tying parameters associated with highly correlated features, we achieve significant reduction of model complexity, with a slight or even no loss in generalization accuracy. We evaluate the proposed method on both a fully connected neural network and a deep convolutional neural network. The results show that GrOWL can compress large DNNs by factors ranging from 11.4 to 14.5, with negligible loss on accuracy.

The correlation patterns identified by GrOWL are close to those of the input features to each layer. This may be important to reveal the structure of the features, contributing to the interpretability of deep learning models. On the other hand, by automatically tying together the parameters corresponding to highly correlated features, GrOWL alleviates the negative effect of strong correlations that might be induced by the noisy input or the co-adaption tendency of DNNs.

The gap in the accuracy vs memory trade-off obtained by applying GrOWL and group-Lasso decreases as we move to large DNNs. Although we suspect this can be caused by running a much larger network on a simple dataset, it motivates us to explore different ways to apply GrOWL to compress neural networks. One possible approach is to apply GrOWL within each neuron by predefining each 2D convolutional filter as a group (instead all 2D convolutional filters corresponding to the same input features). By doing so, we encourage parameter sharing among much smaller units, which in turn would further improve the diversity vs parameter sharing trade-off. We leave this for future work.

---

[4]Although parameter sharing is not considered by Li et al. (2016), according to Table 2, pruning following weight decay together with parameter sharing still cannot compress the network as much as GrOWL does.

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

## APPENDIX A  PROXGROWL

Various methods have been proposed to compute the proximal mapping of OWL (ProxOWL) . It has been proven that the computation complexity of these methods is $O(n \log n)$ which is just slightly worse than the soft thresholding method for solving $\ell_1$ norm regularization. In this paper, we use Algorithm 2 that was originally proposed in Bogdan et al. (2015).

---

**Algorithm 2** ProxGrOWL Bogdan et al. (2015) for solving $\text{prox}_{\eta, \Omega_\lambda}(z)$

---

**Input**: $z$ and $\lambda$
Let $\widetilde{\lambda} = \eta\lambda$ and $\widetilde{z} = |Pz|$ be a nonincreasing vector, where $P$ is a permutation matrix.
**while** $\widetilde{z} - \widetilde{\lambda}$ is not nonincreasing: **do**
  Identify strictly increasing subsequences, *i.e.,* segments $i : j$ such that

$$\widetilde{z}_i - \widetilde{\lambda}_i < \widetilde{z}_{i+1} - \widetilde{\lambda}_{i+1} < \widetilde{z}_j - \widetilde{\lambda}_j \tag{12}$$

  Replace the values of $\widetilde{z}$ and $\widetilde{\lambda}$ over such segments by their average value:  for $k \in \{i, i+1, \cdots, j\}$

$$\widetilde{z}_k \leftarrow \frac{1}{j-i+1} \sum_{i \le k \le j} \widetilde{z}_k, \qquad \widetilde{\lambda}_k \leftarrow \frac{1}{j-i+1} \sum_{i \le k \le j} \widetilde{\lambda}_k \tag{13}$$

**end while**
**Output:** $\widehat{z} = \text{sign}(z) P^T (\widetilde{z} - \widetilde{\lambda})_+$.

---

## APPENDIX B  AFFINITY PROPAGATION

Affinity Propagation is a clustering method based on sending messages between pairs of data samples. The idea is to use these messages to determine the most representative data samples, which are called exemplars, then create clusters using these exemplars.

Provided with the precomputed *data similarity* $s(i, j), i \ne j$ and *preference* $s(i, i)$, there are two types information being sent between samples iteratively: 1) *responsibility* $r(i, k)$, which measures how likely that sample $k$ should be the exemplar of sample $i$; 2) *availability* $a(k, i)$, which is the evidence that sample $i$ should choose sample $k$ as its exemplar. The algorithm is described in 3.

---

**Algorithm 3** Affinity Propagation Frey & Dueck (2007)

---

**Initialization**: $r(i, k) = 0, a(k, i) = 0$ for all $i, k$
**while** not converge **do**
  **Responsibility updates:**

$$r(i, k) \leftarrow s(i, k) - \max_{j \ne k}(a(j, i) + s(i, j))$$

  **Availability updates:**

$$a(k, k) \leftarrow \sum_{j \ne k} \max\{0, r(j, k)\}$$

$$a(k, i) \leftarrow \min\left(0, r(k, k) + \sum_{j \notin \{k, i\}} \max\{0, r(j, k)\}\right)$$

**end while**
**Making assignments:**

$$c_i^* \leftarrow \arg\max_k r(i, k) + a(k, i)$$

---

Unlike k-means or agglomerative algorithm, Affinity Propagation does not require the number of clusters as an input. We deem this as a desired property for enforcing parameter sharing in neural network compression because it's impossible to have the exact number of clusters as a prior information. In practice, the input *preference* of Affinity Propagation determines how likely each sample will be chosen as an exemplar and its value will influence the number of clusters created.

## APPENDIX C    VGG-16 ON CIFAR-10

Table 3: Network statistics of VGG-16.

| Layers | Output $w \times h$ | #Channels in&out | #Params |
|---|---|---|---|
| conv1 | $32 \times 32$ | 3,  64 | 1.7E+03 |
| *conv2 | $32 \times 32$ | *64*,  64 | 3.7E+04 |
| *conv3 | $16 \times 16$ | *64*,  128 | 7.4E+04 |
| *conv4 | $16 \times 16$ | *128*,  128 | 1.5E+05 |
| *conv5 | $8 \times 8$ | *128*,  128 | 2.9E+05 |
| *conv6 | $8 \times 8$ | *128*,  256 | 5.9E+05 |
| *conv7 | $8 \times 8$ | *256*,  256 | 5.9E+05 |
| *conv8 | $4 \times 4$ | *256*,  512 | 1.2E+06 |
| *conv9 | $4 \times 4$ | *512*,  512 | 2.4E+06 |
| *conv10 | $4 \times 4$ | *512*,  512 | 2.4E+06 |
| *conv11 | $2 \times 2$ | *512*,  512 | 2.4E+06 |
| *conv12 | $2 \times 2$ | *512*,  512 | 2.4E+06 |
| *conv13 | $2 \times 2$ | *512*,  512 | 2.4E+06 |
| *fc | 1 | *512*,  512 | 1.0E+06 |
| sofrmax | 1 | 512,  10 | 5.1E+03 |

Table 4: VGG: Clustering rows over different preference values for running the *affinity propagation algorithm* (Algorithm 3). For each experiment, we report clustering accuracy (**A**), compression rate (**C**), and parameter sharing (**S**) of layers 9-14. For each regularizer, we use different preference values to run Algorithm 3 to cluster the rows at the end of initial training process. Then we retrain the neural network correspondingly. The results are reported as the averages over 5 training and retraining runs.

| Preference Value | 0.6 (A, C, S) | 0.7 (A, C, S) | 0.8 (A, C, S) | 0.9 (A, C, S) |
|---|---|---|---|---|
| GrOWL | 92.2%, 13.6, 3.5 | 92.2%, 12.5, 2.6 | 92.2%, 11.4, 2.1 | 92.2%, 10.9, 1.7 |
| Group Lasso | 92.2%, 12.1, 1.1 | 92.0%, 11.4, 1.1 | 92.1%, 11.0, 1.0 | 92.2%, 9.5, 1.0 |
| GrOWL + $\ell_2$ | 92.7%, 14.7, 2.3 | 92.5%, 15.4, 2.9 | 92.7%, 14.5, 2.3 | 92.6,%, 13.5, 1.8 |
| GrLasso + $\ell_2$ | 92.7%, 14.8, 1.2 | 92.7%, 14.5, 1.1 | 92.7%, 14.5, 1.0 | 92.6%, 14.3, 1.0 |
| Weight Decay | 93.2%, 1.8, 2.2 | 93.4%, 1.5, 1.7 | 93.1%, 1.3, 1.4 | 93.3%, 1.1, 1.1 |

