# OpenReview forum: "LEARNING TO SHARE: SIMULTANEOUS PARAMETER TYING AND SPARSIFICATION IN DEEP LEARNING"
_ICLR.cc/2018/Conference — Accept (Poster)_

### Official Review · AnonReviewer3 · 2017-11-26
**Unclear motivation and insufficient experimental results**

**Rating:** 6
**Confidence:** 3

**Review:**

This paper proposes to apply a group ordered weighted l1 (GrOWL) regularization term to promote sparsity and parameter sharing in training deep neural networks and hence compress the model to a light version.

The GrOWL regularizer (Oswal et al., 2016) penalizes the sorted l2 norms of the rows in a parameter matrix with corresponding ordered regularization strength and the effect is similar to the OSCAR (Bondell & Reich, 2008) method that encouraging similar (rows of) features to be grouped together.

A two-step method is used that the regularizer is applied to a deep neural network at the initial training phase, and after obtaining the parameters, a clustering method is then adopted to force similar parameters to share the same values and then the compacted neural network is retrained. The major concern is that a much more complicated neural network (with regularizations) has already been trained and stored to obtain the uncompressed parameters. What’s the benefit of the compression and retraining the trained neural network?

In the experiments, the performance of the uncompressed neural network should be evaluated to see how much accuracy loss the regularized methods have. Moreover, since the compressed network loses accuracy, will a smaller neural network can actually achieve similar performance compared to the compressed network from a larger network? If so, one can directly train a smaller network (with similar number of parameters as the compressed network) instead of using a complex two-step method, because the two-step method has to train the original larger network at the first step.

---

> ### Author Response · Authors · 2018-01-05
> **Motivation for compressing the deep neural network and why we try to compress a large network instead of directly training a small one.**
>
> We'd like to thank the reviewer for the questions. Below we further clarify our motivation and answer the questions raised.
>
> (1) We start by addressing the reviewer's major concern about the motivation for compressing the DNN and why we try to compress a large network instead of directly training a small one.
>
> a) A modern DNN is typically very large, which results in a heavy burden on both computation and memory. Therefore, it’s very hard to deploy such large DNNs in embedded sensors or mobile devices, where computational and memory resources are scarce. This motivates working on DNN compression without significantly hurting the performance. Some approaches have been proposed in recent years: for example, Han et al  [1] have shown that, after compression, a large DNN, such as AlexNet and VGGNet, can be stored in on-chip SRAM instead of off-chip DRAM. Please refer to the related work section of our paper for details.
>
> (b) Researchers have experienced difficulties in training small networks to achieve good generalization performance [2], this being one of the main reasons why DNNs gained popularity. Starting with a large NN, our compression method can be understood as automatically finding a small network that achieves comparable performance as a large one. Without this compression process, it will be very hard to achieve similar performance as the large network when directly training a small NN from scratch.
>
> To further demonstrate this claim, we can take the CIFAR-10 as an example. The best performance obtained by running VGG on CIFAR-10 without compression is 93.4% (Table 2 in the paper). By using GrOWL with L2 or group-Lasso plus L2, we can compress it by almost 15X, achieving 92.7%  accuracy. The total number of free parameters of the compressed network is 1.03e+06, versus 1.5E+07 of the original uncompressed VGG network. However, the best accuracy achieved by a small network (www.tensorflow.org/tutorials/deep_cnn) with a similar number of parameters (1.06E+06) is 86%.
>
> (2) Concerning the question of why we retrain the trained network, there are two main reasons for the proposed training-retraining pipeline.
>
> a) Debiasing: it is well-known that L1-type regularization not only prunes away redundant parameters but also shrinks the surviving ones towards zero, thus yielding biased estimates thereof, which can harm the network’s performance. The retraining process helps to reduce this bias.
>
> b) Enforcing the learned tying structure associated with strongly correlated features: unlike standard sparsity-inducing regularizers (Lasso or group-Lasso), GrOWL not only eliminates unimportant parameters (neurons) by setting all the corresponding weights to zero but it also explicitly identifies strongly correlated neurons by tying the corresponding weights to be very close to each other or even exactly equal. This ability of GrOWL motivates the following two-stage procedure: (i) use GrOWL regularization in the training process to simultaneously identify significant neurons and groups of parameter that should be tied together; (ii) retrain the network, enforcing the structure that was unveiled in the previous phase, i.e., keeping only the significant neurons and enforcing the learned tying structure.
>
> The strongly correlated features of a DNN can be generated by the noisy inputs or by the co-adaption tendency of DNN training [3]. Therefore, by enforcing parameter sharing among those parameters that correspond to strongly correlated features, we expect to alleviate the negative effects caused by either noisy input or co-adaption.
>
> (3) We have added the performance of uncompressed neural networks for both MNIST and CIFAR-10 experiment.
>
>
> [1] S. Han, H. Mao, and W. Dally. "Deep compression: Compressing deep neural networks with pruning, trained quantization and Huffman coding", International Conference on Learning Representations, 2016.
>
> [2] L. J. Ba and R. Caruana. "Do deep nets really need to be deep?" NIPS 2014.
>
> [3] N. Srivastava, G. E Hinton, A. Krizhevsky, I. Sutskever, and R. Salakhutdinov. "Dropout: a simple way to prevent neural networks from overfitting", Journal of Machine Learning Research,  vol. 15, pp. 1929−1958, 2014.

---

> > ### Comment · AnonReviewer3 · 2018-01-14
> > **Response to authors' rebuttal**
> >
> > Thank the authors for their detailed response to my questions. The revision and response provided clearer explantation for the motivation of compressing a deep neural network. Additional experimental results were also included for the uncompressed neural net. I would like to change my rating based on these updates.

---

> ### Comment · AnonReviewer1 · 2018-01-10
> **Compression vs smaller networks**
>
> While it may appear a bit like black magic, it is empirically true that often training a large network and compressing it will outperform a small network of the same size trained on the same data. The reference to Han et al. cited by the authors above explains this in more detail.

---

> > ### Comment · AnonReviewer3 · 2018-01-14
> > **Thanks for the comments.**
> >
> > Thanks for the comments.

---

### Official Review · AnonReviewer2 · 2017-11-28
**A nice paper that would be more compelling with a comparison with the group elastic net**

**Rating:** 8
**Confidence:** 5

**Review:**

SUMMARY
The paper proposes to apply GrOWL regularization to the tensors of parameters between each pair of layers. The groups are composed of all coefficients associated to inputs coming from the same neuron in the previous layer. The proposed algorithm is a simple proximal gradient algorithm using the proximal operator of the GrOWL norm. Given that the GrOWL norm tend to empirically reinforce a natural clustering of the vectors of coefficients which occurs in some layers, the paper proposes to cluster the corresponding parameter vectors, to replace them with their centroid and to retrain with the constrain that some vectors are now equal. Experiments show that some sparsity is obtained by the model and that together with the clustering and high compression of the model is obtained which maintaining or improving over a good level of generalization accuracy. In comparison, plain group Lasso yields compressed versions that are too sparse, and tend to degrade performance. The method is also competitive with weight decay with much better compression.

REVIEW
Given the well known issue that the Lasso tends to select arbitrarily and in a non stable way variables
that are correlated *but* given that the well known elastic-net (and conceptually simpler than GrOWL) was proposed to address that issue already more than 10 years ago, it would seem relevant to compare the proposed method with the group elastic-net.

The proposed algorithm is a simple proximal gradient algorithm, but since the objective is non-convex it would be relevant to provide references for convergence guarantees of the algorithm.

How should the step size eta be chosen? I don't see that this is discussed in the paper.

In the clustering algorithm how is the threshold value chosen?

Is it chosen by cross validation?

Is the performance better with clustering or without?

Is the same threshold chosen for GrOWL and the Lasso?

It would be useful to know which values of p, Lambda_1 and Lambda_2 are selected in the experiments?

For Figures 5,7,8,9 given that the matrices do not have particular structures that need to be visualized but that the important thing to compare is the distribution of correlation between pairs, these figures that are hard to read and compare would be advantageously replaced by histograms of the values of the correlations between pairs (of different variables). Indeed, right now one must rely on comparison of shades of colors in the thin lines that display correlation and it is really difficult to appreciate how much of correlation of what level are present in each Figure. Histograms would extract exactly the relevant information...

A brief description of affinity propagation, if only in the appendix, would be relevant.
Why this method as opposed to more classical agglomerative clustering?

A brief reminder of what the principle of weight decay is would also be relevant for the paper to more self contained.

The proposed experiments are compelling, except for the fact that it would be nice to have a comparison with the group elastic-net.

I liked figure 6.d and would vote for inclusion in the main paper.


TYPOS etc

3rd last line of sec. 3.2 can fail at selecting -> fail to select

In eq. (5) theta^t should be theta^{(t)}

In section 4.1 you that the network has a single fully connected layer of hidden units -> what you mean is that the network has a single hidden layer, which is furthermore fully connected.

You cite several times Sergey (2015) in section 4.2. It seems you have exchanged first name and last name plus the corresponding reference is quite strange.

Appendix B line 5 ", while." -> incomplete sentence.

---

> ### Author Response · Authors · 2018-01-05
> **Adding comparison with group elastic net**
>
> We sincerely thank the reviewer for the thoughtful comments and suggestions. In order to address the main concern, we updated our paper by including a comparison with group elastic net (group-EN). Below, we answer the questions one by one.
>
> (1)  We agree that a comparison between GrOWL and group-EN (group-Lasso + L2) improves the paper. We updated the results for both MNIST and CIFAR10; we found that by applying either GrOWL or group Lasso with L2 regularization, the accuracy and memory trade-off improve. We didn’t observe any significant difference between the compression vs accuracy trade-off achieved by GrOWL + L2 and group-EN. However, GrOWL alone still yields a more obvious parameter sharing effect (Tables 1, 2, and 4) than group-EN. We suspect this is due to the inability of L2 + L1 regularization to encourage group-level parameter sharing. As seen in Tables 2 and 4, applying group-Lasso, with or without L2 regularization, doesn’t make a significant difference in parameter sharing, which may imply that, at a group level, both group-Lasso and group-EN tend to identify only a subset of the correlated features.
>
> We should also mention that we adjusted the weight of the L2 norm in group-EN to yield the best compression vs accuracy trade-off. Although the parameter sharing effect for group-EN is improved slightly by using large weight for the L2 regularizer, the performance degrades as a consequence.
>
> (2)  In order to identify more correlations, we choose large lambda_2 (ranging from 0.05 to 0.1). Preventing the zeroing out of all the parameters is achieved by using a relatively small step size for all the networks. We use step sizes of 1e-3 and 1e-2 for the fully connected neural network (NN) on MNIST and the VGG on CIFAR-10, respectively. We included the choices of step size in the corresponding section of the paper.
>
> (3) By using a comparatively larger preference value, we only cluster identical or very similar rows together and enforce parameter sharing therein. Consequently, those regularizers that do not identify correlations in the same way, including group-Lasso and group-EN, almost no parameter sharing (see Table 2) is enforced and the retraining process serves essentially as a debiasing phase, which helps to improve the generalization performance. On the other hand, although it has been proved in the linear regression case (Theorem 1 in [1]) that GrOWL only yields strictly equal rows if they correspond to identical features, we didn’t observe any significant difference in accuracy between retraining with and without enforcing parameter sharing. This justifies our motivation to enforce parameter sharing only among the parameter groups that correspond to strongly correlated input features. However, for GrOWL, we did expect an improvement by enforcing parameter sharing among those parameters that correspond to strongly correlated features in the retraining process. Our intuition is that the sharing (averaging) process alleviates the negative effects caused by the noisy inputs or the co-adaptation effect. Further exploring the reasons underlying the absence of improvement when retraining while enforcing parameter sharing in this scenario is an interesting direction for future work.
>
> (4)  We run the clustering algorithm over different preference values and choose the one that works well for all of the regularizers. We use preference value 0.6 for MNIST and 0.8 for CIFAR-10. We provide many more details about the compression and accuracy trade-off in Table 4 (Appendix D).
>
> (5)  In the revised manuscript, we provide a brief description of the affinity propagation (AP) algorithm (Appendix B). As for the why we chose AP instead of classical agglomerative clustering, the reason was that it does not require setting the number of clusters a priori.
>
> We provide the p values in our paper. We used comparatively large lambda_2 (ranging from 0.05 to 0.1) to identify correlations, and lambda_1 is chosen to balance the sparsity and accuracy trade-off.  We will provide more detailed information after we clean up the code and release it.
>
> (6)  We briefly discuss weight decay at the end of section 4.1.
>
> (7)  Thanks for your suggestion, we already move Fig. 6.d and the corresponding section of the main body of the paper.
>
> [1] U. Oswal, C. Cox, M. Lambon-Ralph, T. Rogers, and R. Nowak. "Representational similarity learning with application to brain networks", Proceedings of The 33rd International Conference on Machine Learning, pp. 1041–1049, 2016.

---

> > ### Comment · AnonReviewer2 · 2018-01-12
> > **Response to rebuttal**
> >
> > I would like to thank the authors for detailed responses to my comment. These responses and the updated version of the paper are compelling. I am therefore updating my rating.
> >
> > I am just confused by point (5): classical agglomerative clustering does not require to choose the number of clusters a priori (unlike k-means for example). The justification for the use of AP is still unclear to me. But this is a minor point.

---

### Official Review · AnonReviewer1 · 2017-12-01
**An incremental improvement for compressing deep neural networks**

**Rating:** 7
**Confidence:** 4

**Review:**

The authors propose to use the group ordered weighted l1 regulariser (GrOWL) combined with clustering of correlated features to select and tie parameters, leading to a sparser representation with a reduced parameter space. They apply the proposed method two well-known benchmark datasets under a fully connected and a convolutional neural network, and demonstrate that in the former case a slight improvement in accuracy can be achieved, while in the latter, the method performs similar to the group-lasso, but at a reduced computational cost for classifying new images due to increased compression of the network.

The paper is well written and motivated, and the idea seems fairly original, although the regularisation approach itself is not new. Like many new approaches in this field, it is hard to judge from this paper and its two applications alone whether the approach will lead to significant benefits in general, but it certainly seems promising.

Positive points:
- Demonstrated improved compression with similar performance to the standard weighted decay method.
- Introduced a regularization technique that had not been previously used in this field, and that improves on the group lasso in terms of compression, without apparent loss of accuracy.
- Applied an efficient proximal gradient algorithm to train the model.

Negative points:
- The method is sold as inducing a clustering, but actually, the clustering is a separate step, and the choice of clustering algorithm might well have an influence on the results. It would have been good to see more discussion or exploration of this. I would not claim that, for example, the fused lasso is a clustering algorithm for regression coefficients, even though it demonstrably sets some coefficients to the same value, so it seems wrong to imply the same for GrOWL.
- In the example applications, it is not clear how the accuracy was obtained (held-out test set? cross-validation?), and it would have been good to get an estimate for the variance of this quantity, to see if the differences between methods are actually meaningful (I suspect not). Also, why is the first example reporting accuracy, but the second example reports error?
- There is a slight contradiction in the abstract, in that the method is introduced as guarding against overfitting, but then the last line states that there is "slight or even no loss on generalization performance". Surely, if we reduce overfitting, then by definition there would have to be an improvement in generalization performance, so should we draw the conclusion that the method has not actually been demonstrated to reduce overfitting?

Minor point:
- p.5, in the definition of prox_Q(epsilon), the subscript for the argmin should be nu, not theta.

---

> ### Author Response · Authors · 2018-01-05
> **Updating all results with averaged accuracy and the corresponding variance**
>
> We sincerely thank the reviewer for the positive feedback as well as the thoughtful and constructive comments. In order to address the main concerns, we updated our paper with the averaged results and the corresponding variances.
>
> (1) The clustering property of GrOWL is claimed as the ability to identify strong correlations among input features, by tying the corresponding weights/parameters to very close values (or to a common value for very strong correlations). In other words, GrOWL identifies clusters associated with correlated features, while a separate clustering algorithm is used to find the clusters formed by GrOWL. We modified our paper to clarify this.
>
> It is true that the clustering algorithm may influence the results. However, in practice, we find the adopted clustering algorithm to be quite stable when used with relatively large threshold values to cluster the rows at the end of the initial training process (since our motivation is to only encourage parameter sharing among very similar rows, which correspond to highly correlated features). We also explore how the threshold values for the clustering algorithm can influence the clustering results in Table 4 (Appendix D).
>
> (2) Thank you for the suggestions; we have modified our paper accordingly.
>
> (i) We have updated all the results with average accuracies and the corresponding variance (Tables 1, 2, and 4). The reviewer is right: the difference in accuracy and memory trade-off does shrink in terms of average results (previously we reported the best result). However, as seen in Table 1, GrOWL still achieves better accuracy vs memory trade-off on MNIST, while for VGG-16 on CIFAR, the improvement is not obvious. We suspect the absence of improvement in this scenario can be due to the following two reasons:
>
>          a) the CIFAR-10 dataset is simple, so that good performance can still be obtained after strong compression of large DNNs (VGG-16 in this paper). We believe this gap in the compression vs accuracy trade-off can be further increased in larger networks on more complex datasets.
>
>         b) parameter sharing is performed among rows of the (reshaped) weight matrix, which might prevent the network from gaining more diversity. One possibility is to apply GrOWL within each neuron, by predefining each 2D convolution filter as a group (instead of all 2D filters corresponding to the same input features), so that within each neuron, the 2D filters that correspond to strongly correlated features will be tied together. By doing so, GrOWL can still identify the correlated features by tying the corresponding filters together, but more diversity of the neural network can be achieved since, for each neuron, the correlations between the input features can be different.
>
> We leave the above two directions for future work.
>
> (ii) Also, in order to be consistent with each other, we report the testing accuracy for both MNIST and CIFAR10. The accuracies in this paper are reported with respect to a held-out test set.
>
> (3) Thank you for pointing out the contradiction. In the previous version, the conclusion “slight or even no loss on generalization performance” was drawn by comparing the accuracy obtained by applying GrOWL to train the neural network to the best accuracy on the same network achieved by the regularizers considered in this paper. We apologize for not making it clear; we corrected this in the revised abstract.

---

### Author Response · Authors · 2018-01-05
**Revision**

(1) We add comparison with group Elastic Net.

(2) We update our results with the averaged ones and the corresponding variances.

---

### Decision · Program_Chairs · 2018-01-29
**ICLR 2018 Conference Acceptance Decision**

**Decision:**

Accept (Poster)

**Comment:**

The paper proposes to regularize via a family of structured sparsity norms on the weights of a deep network.  A proximal algorithm is employed for optimization, and results are shown on synthetic data, MNIST, and CIFAR10.

Pros: the regularization scheme is reasonably general, the optimization is principled, the presentation is reasonable, and all three reviewers recommend acceptance.

Cons: the regularization is conceptually not terribly different from other kinds of regularization proposed in the literature.  The experiments are limited to quite simple data sets.